# Lung Ultrasound Assessment of Lung Injury Following Acute Spinal Cord Injury in Rats

**DOI:** 10.3390/diagnostics15202648

**Published:** 2025-10-21

**Authors:** Na Ni, Ruiliang Chu, Kai Gu, Yi Zhong

**Affiliations:** 1Department of Pediatrics, Women and Children’s Hospital of Chongqing Medical University, Department of Pediatrics, Chongqing Health Center for Women and Children, Chongqing 400000, China; bei-chen@163.com; 2Department of Ultrasound Children’s Hospital of Chongqing Medical University, National Clinical Research Center for Child Health and Disorders, Ministry of Education Key Laboratory of Child Development and Disorders, Chongqing Engineering Research Center of Stem Cell Therapy, Chongqing 400000, China; dcrl987@126.com (R.C.); 482011@hospital.cqmu.edu.cn (K.G.)

**Keywords:** lung ultrasound, acute spinal cord injury, pulmonary edema, B-line score, respiratory function, pulmonary pathology, rodent model, non-invasive imaging

## Abstract

**Background/Objectives:** Acute spinal cord injury (ASCI) often leads to pulmonary complications, yet reliable, non-invasive assessment tools are limited. This study aimed to evaluate the utility of lung ultrasound (LUS) in assessing lung injury following ASCI in a rat model. **Methods:** Fifty-four female Sprague Dawley rats were randomized into sham (*n* = 27) or ASCI (*n* = 27) groups. LUS was performed at 12 h, 48 h, and 1 week post-injury, with lung injury quantified using a modified B-line score (BLS). Pulmonary function was assessed non-invasively, and histopathological evaluation and wet-to-dry (W/D) weight ratios were conducted post-mortem. Correlations between BLS and functional and pathological parameters were analyzed. **Results:** Histological analysis revealed progressive pulmonary hemorrhage, edema, and inflammatory infiltration peaking at 48 h post-injury, with residual hemorrhage and fibroplasia at 1 week. LUS findings evolved from narrow-based B-lines at 12 h to confluent B-lines with pleural abnormalities by 1 week. ASCI rats showed significant reductions in respiratory frequency, peak inspiratory and expiratory flow, and EF_50_ at all time points (*p* < 0.05). Tidal volume and minute volume decreased initially, with partial recovery at 1 week. BLS negatively correlated with all pulmonary function parameters and positively with the histological score and W/D ratio (*p* < 0.001). **Conclusions:** LUS reliably detects and tracks lung injury after ASCI, correlating well with physiological and pathological indicators. These findings support its potential as a non-invasive monitoring tool. Future refinement of ultrasound scoring may improve clinical applicability in ASCI-related pulmonary assessment.

## 1. Introduction

Acute spinal cord injury (ASCI) is a devastating traumatic event that presents serious challenges to patient survival and long-term health. Among the complications following acute spinal cord injury (ASCI), pulmonary dysfunction remains a leading cause of morbidity and mortality. Notably, studies have demonstrated that the lungs are the most frequently injured visceral organs across all levels of ASCI (cervical, thoracic, or lumbar), with more than 25% of patients experiencing lung complications [1]. Prompt airway management has been shown to reduce the incidence of these complications [2]. However, respiratory support techniques, especially mechanical ventilation, are associated with increased risks, such as ventilator-associated pneumonia and secondary infections [3].

Accurate and timely assessment of pulmonary injury is therefore critical in guiding respiratory care in ASCI patients. Computed tomography (CT), while capable of delivering high-resolution diagnostic images [4], presents several limitations in this setting. Transporting unstable patients for imaging increases the risk of exacerbating spinal injuries, and repeated CT scans involve cumulative radiation exposure, which is particularly problematic in the acute phase [5]. Rebecca et al. reported that excessive exposure to CT radiation significantly increases the risk of developing hematological malignancies [6]. In contrast, lung ultrasound (LUS) is a non-invasive, radiation-free, and bedside-compatible imaging modality that enables dynamic evaluation of lung conditions. LUS relies on the generation of reflective artifacts, such as B-lines, based on the air-to-fluid ratio in lung tissue, allowing for semi-quantitative assessment of extravascular lung water. This technique has been validated in animal studies as a reliable method for detecting pulmonary edema [7,8]. Hao et al. demonstrated that LUS scoring effectively predicts pulmonary complications following blunt thoracic trauma [9]. Additionally, He et al. and our prior research confirmed that pulmonary edema and hemorrhage frequently occur during the early phase of ASCI [10,11]. Importantly, our findings suggest that lung injury after ASCI is a complex and multifactorial pathological process, not limited to edema and hemorrhage. This distinction implies that ASCI-related lung injury may differ fundamentally from typical post-traumatic pulmonary contusions or acute respiratory distress syndrome (ARDS). Therefore, there is a pressing need to better characterize this distinct pathological entity and develop appropriate diagnostic tools.

This study aims to investigate the utility of lung ultrasound for dynamically assessing lung injury following ASCI in a rat model. By integrating physiological measurements with pathological evaluations, we seek to clarify the value of LUS as a real-time, non-invasive modality for pulmonary assessment and to lay the groundwork for improved clinical monitoring in ASCI.

## 2. Materials and Methods

### 2.1. Animals

A total of 54 female, wild-type, specific pathogen-free (SPF) grade Sprague Dawley rats (6–7 week, 220–250 g) were used in this study. Animals were randomly allocated using a random number table into two groups: a sham-operated control group (*n* = 27) and an acute spinal cord injury (ASCI) group (*n* = 27). At each designated time point (12 h, 48 h, and 1 week post-treatment), nine rats from each group were humanely euthanized for analysis. Rats were housed under standard environmental conditions (temperature: 24–26 °C; relative humidity: 50–55%; 12 h/12 h light/dark cycle). All animal procedures were approved by the Association for Assessment and Accreditation of Laboratory Animal Care (Chongqing, China) and the Animal Ethics Committee of Chongqing Medical University (Approval Nos. SCXK[Yu]2022-0010 and SYXK[Yu]2022-0016). The ASCI model was established using a modified Allen’s method [12], with an impact energy of 120 g·cm at T10 vertebral level spinal cord. Anesthesia was induced using sodium pentobarbital (40 mg/kg, intraperitoneal; Sigma-Aldrich, St. Louis, MO, USA) to minimize pain.

### 2.2. Lung Ultrasound

Lung ultrasound (LUS) examinations were conducted using two systems: the Mindray M9 (Mindray Medical, Shenzhen, China) equipped with a 12–4 MHz L12-4 linear array probe, and the Philips EPIQ Elite (Philips Healthcare, Andover, MA, USA) equipped with an 18–4 MHz L18-4 linear array probe. All scans were performed in B-mode using standardized settings: focal point at the pleural line, fundamental imaging mode enabled, and spatial compound imaging disabled to optimize visualization of pleural structures, A-lines, and B-lines. LUS was performed independently by two experienced ultrasound physicians. The two ultrasound systems were randomly assigned for use through a randomization scale method. The thorax was divided into four scanning regions (two anterior, two posterior), covering 5–6 intercostal spaces per region to ensure full pulmonary field assessment [13]. The probe was placed perpendicular to the body surface. In normal aerated lungs, A-lines (horizontal, repetitive reverberation artifacts parallel to the pleural line) were considered a baseline pattern. B-lines were defined as vertical, laser-like hyperechoic artifacts arising from the pleural line, extending to the edge of the screen without fading, and moving synchronously with lung sliding. Lung injury severity was quantified using a modified B-line scoring (BLS) system [8]. A score of 0 indicated normal A-lines only. A score of 1 represented a single narrow-based B-line. Multiple B-lines with partial coalescence or wide-based spacing received a score of 2. Finally, a score of 3 was assigned for confluent B-lines or other significant pulmonary abnormalities. Each area has a maximum score of 3 points, and the results are obtained by summing the scores of each area. The average score from the evaluations of two ultrasound doctors is taken.

### 2.3. Pulmonary Edema Evaluation

Pulmonary edema was assessed via the wet-to-dry (W/D) lung weight ratio. After euthanasia, the lungs were excised and separated into left and right lobes. Wet weights were recorded immediately, followed by drying at 60 °C for 48 h to obtain dry weights. The W/D ratio was then calculated to evaluate extravascular lung water content.

### 2.4. Histopathological Examination

To preserve pulmonary structure for histological analysis, 40 mL of normal saline followed by 40 mL of 4% paraformaldehyde were instilled into the trachea. Lungs were then removed from the thoracic cavity and fixed in 4% paraformaldehyde for 72 h. Fixed tissues were embedded in paraffin, sectioned at 4 μm thickness, and stained with hematoxylin and eosin (H&E) for light microscopy. Histologic scoring was performed under high-power (200×) magnification at three randomly selected sites. The pathologist was blinded to group assignments (injury vs. ASCI). Scoring assessed categories including inflammatory cell infiltration, edema, congestion, and intra-alveolar hemorrhage. Scores were defined as follows: 0 = normal; 1 = mild injury; 2 = moderate injury; 3 = severe injury, yielding a maximum possible score of 12 [10].

### 2.5. Lung Function Tests 

Pulmonary function was assessed using a ventilated bias flow whole-body plethysmography (WBP) system (FinePointe WBP, DSI, New Brighton, MN, USA). Prior to testing, rats were acclimatized for 1 h. Each test session included a 7 min recording period to capture stable breathing patterns. The following parameters were recorded: respiratory frequency (ƒ, breaths/min), tidal volume (V_T_, mL), minute volume (MV, mL/min), peak expiratory flow (PEF, mL/s), and tidal mid-expiratory flow (EF_50_, mL/s) [14].

### 2.6. Statistical Analysis

All data are presented as mean ± standard deviation (SD). One-way analysis of variance (ANOVA) was used to evaluate differences among groups at each time point, followed by least significant difference (LSD) post hoc tests for pairwise comparisons between sham and ASCI groups at 12 h, 48 h, and 1 week. Spearman’s rank correlation coefficients (r) were calculated to assess associations between BLS and pulmonary function parameters, histological scores, and W/D ratios. Analyses were conducted using SPSS software version 22.0 (IBM Corp., Armonk, NY, USA). Statistical significance was defined as *p* < 0.05.

## 3. Results

### 3.1. Gross and Histopathological Evaluation of the Lung Following ASCI

Lungs were collected at defined time points after ASCI and examined macroscopically and histologically (Figure 1a,b). In the ASCI group, patchy pulmonary hemorrhage was visible as early as 12 h post-injury and became more extensive by 48 h, involving all lobes. At 1 week, hemorrhagic areas decreased but persisted focally, accompanied by fibrous tissue proliferation and pleural adhesions. Histological examination revealed normal alveolar architecture in sham-operated rats. In contrast, injured lungs exhibited progressive pathological changes over time: mild hemorrhage, interstitial widening, and inflammatory cell infiltration at 12 h; marked alveolar collapse, inflammatory infiltration, and structural destruction by 48 h; and localized residual hemorrhage with fibrotic changes at 1 week.

### 3.2. Histological Score and Pulmonary Edema of Lung

Quantitative analysis of histological scores (Figure 2a) demonstrated a statistically significant increase in lung injury severity at all post-ASCI time points compared to the sham group (*p* < 0.05). Pulmonary edema was evaluated using the wet-to-dry (W/D) lung weight ratio (Figure 2b), which also showed significantly elevated values at 12 h, 48 h, and 1 week after injury relative to controls (*p* < 0.05).

### 3.3. Lung Ultrasound and B-Line Scoring

LUS imaging performed at each time point revealed a clear progression of lung injury features (Figure 3a–d). Sham rats showed only A-lines, indicating normal lung aeration. In contrast, ASCI rats developed narrow-based B-lines at 12 h, which evolved into confluent, wide-based B-lines by 48 h. At 1 week, persistent B-line fusion and moth-eaten erosive pleural changes were observed. B-line scores (BLS) increased significantly in the ASCI group compared to the sham group at all examined time points (*p* < 0.05; Figure 3e), indicating progressive extravascular lung water accumulation and structural abnormalities.

### 3.4. Pulmonary Function and Correlation Analysis

Pulmonary function parameters were assessed using whole-body plethysmography (Figure 4a–f). Respiratory frequency (ƒ), peak inspiratory flow (PIF), peak expiratory flow (PEF), and tidal mid-expiratory flow (EF_50_) were significantly reduced in ASCI rats at all time points compared to controls (*p* < 0.05). Tidal volume (V_T_) decreased at 12 and 48 h but increased above sham levels by 1 week (*p* < 0.05). Minute volume (MV) followed a similar pattern: initial decline with no statistically significant difference from sham by 1 week. Correlation analysis revealed that BLS was significantly and negatively correlated with all measured pulmonary function parameters (ƒ, V_T_, MV, PIF, PEF, and EF_50_; all *p* < 0.01), with the strongest associations observed for PIF (r = −0.7566) and EF_50_ (r = −0.8175). Conversely, BLS positively correlated with histological injury scores (r = 0.8387, *p* < 0.001) and W/D ratios (r = 0.7544, *p* < 0.001), as shown in Figure 4g.

## 4. Discussion

This study demonstrates that ASCI induced by hyperimpact in rats results in marked and progressive pulmonary injury. Gross and histological analyses revealed extensive pulmonary hemorrhage and edema as early as 12 h post-injury, with worsening pathology by 48 h, and signs of persistent damage including localized fibrotic proliferation at 1 week. These results are consistent with previous findings from our group [11], although injury severity appeared to peak earlier under hyperimpact conditions. The observed pathological changes resemble those reported in pulmonary contusion and blast injuries [9,15], yet the etiology here is distinct: rather than direct thoracic trauma, lung injury arises secondary to spinal cord insult, as supported by prior studies [10,12]. The progressive development of pulmonary edema, confirmed by elevated wet-to-dry (W/D) ratios, and the emergence of fibrotic features suggest that ASCI-induced lung injury is a complex and evolving pathological process. This complexity challenges current clinical decision-making, as no established criteria guide ventilation strategies or airway management following ASCI. Therefore, real-time, bedside-compatible tools for pulmonary assessment are critically needed. LUS has emerged as a valuable imaging modality in critical care settings due to its non-invasive nature, lack of radiation, and high diagnostic accuracy. Previous studies have demonstrated strong correlations between LUS findings and pulmonary edema in both animal models and human patients [7,8]. For instance, LUS outperforms chest radiography in detecting interstitial syndromes in heart failure, with sensitivity and specificity reaching 98% and 88%, respectively [16]. LUS is also increasingly used in ARDS to optimize ventilatory settings [17], highlighting its clinical utility in diverse pulmonary pathologies.

In our study, LUS effectively tracked the dynamic evolution of lung injury following ASCI. At 12 h, narrow-based B-lines (indicative of early alveolar-interstitial syndrome) were visible. By 48 h, B-lines had become confluent with widened bases, paralleling worsening edema and hemorrhage. At 1 week, B-line fusion persisted, accompanied by moth-eaten pleural changes. These ultrasound features may reflect not only ongoing edema but also structural alterations such as fibrosis. Similar signs have been observed in children with bronchopulmonary dysplasia [18], and may relate to capillary proliferation, fibrous tissue development, or pleural surface degeneration [19]. To account for these structural features, we modified the BLS system by assigning a maximum score of 3 to regions exhibiting pleural moth-eaten changes. This adjustment allowed for semi-quantitative assessment of both fluid and fibrotic contributions to lung injury. Our results showed that BLS correlated positively with histopathological scores and W/D ratios, supporting its validity as an imaging biomarker of pulmonary damage following ASCI.

We also assessed lung function parameters using non-invasive plethysmography. Consistent with previous reports [20], BLS was negatively correlated with respiratory frequency (ƒ), tidal volume (V_T_), minute volume (MV), peak inspiratory and expiratory flows (PIF, PEF), and EF_50_. However, the strength of these correlations was lower than those with histological injury and W/D ratio, suggesting that BLS may more accurately reflect structural than functional impairment, particularly in the chronic phase. By 1 week post-injury, functional parameters began to diverge from the injury pattern suggested by BLS. For example, MV showed no significant difference from sham controls, while V_T_ paradoxically increased, suggesting a compensatory shift toward deeper, slower breathing. These trends imply that pulmonary mechanics are influenced not only by fluid accumulation but also by evolving airway resistance and fibrotic remodeling. This interpretation is supported by prior studies indicating that fibrosis contributes to increased bronchial resistance during the chronic phase of ASCI [21]. Importantly, our findings underscore the limitations of existing BLS systems in detecting fibrotic progression. The moth-eaten pleural sign, while suggestive, lacks standardised scoring criteria. As Zhou et al. have shown, LUS can detect pleural surface changes associated with varying degrees of fibrosis, including alterations in echogenicity, smoothness, and lesion angles [22]. Thus, further refinement of ultrasound criteria is necessary to reliably distinguish fibrotic from edematous injury.

This study provides strong preclinical evidence that LUS is a reliable, practical, and non-invasive tool for dynamically assessing ASCI-induced lung injury. However, several limitations must be acknowledged. First, the sample size was relatively small and limited to a single animal model and lacks inter-rater reliability assessment in BLS evaluation. Second, while the inclusion of sham controls enhances interpretability, clinical extrapolation remains speculative. This study has not been implemented in clinical practice, and future research may be needed to clarify the actual value of LUS in patients with ASCI. Third, the modified BLS system, though promising, requires validation in larger studies and human subjects.

Future research should focus on developing refined ultrasound-based scoring systems that integrate both fluid and fibrotic parameters. Longitudinal studies involving larger animal cohorts and clinical trials will be essential to validate these findings and determine the translational utility of LUS in ASCI management.

## 5. Conclusions

LUS provides a reliable, non-invasive approach for assessing lung injury following ASCI. It effectively reflects both pathological and physiological changes in the acute phase. However, as the complexity of lung injury evolves, particularly with the development of fibrosis, current scoring systems may be insufficient. Further refinement of LUS-based evaluation methods is warranted to improve diagnostic precision and to better guide clinical decision-making in ASCI-related pulmonary management.

## Figures and Tables

**Figure 1 diagnostics-15-02648-f001:**
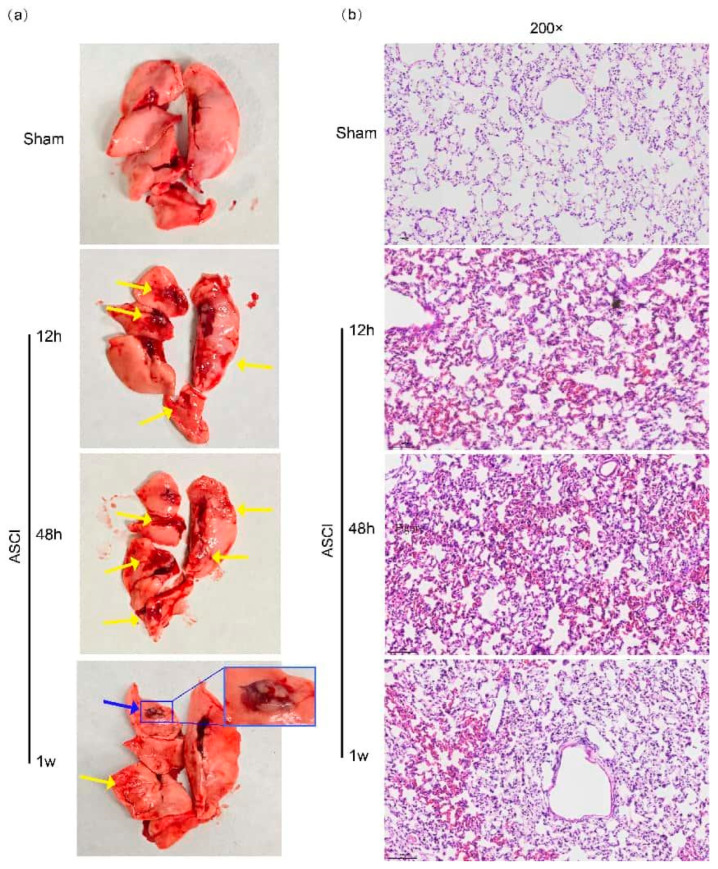
Macroscopic and Histological Changes in Pulmonary Tissue Following Acute Spinal Cord Injury in Rats. Representative gross (**a**) and histopathological (**b**) images of lung tissue at 12 h, 48 h, and 1 week after acute spinal cord injury (ASCI) or sham surgery. (**a**) Gross morphology shows pulmonary hemorrhagic foci (yellow arrows) and regions of fibrotic proliferation with pleural adhesions (blue arrows). (**b**) Hematoxylin and eosin (H&E)-stained lung sections reveal alveolar septal thickening, inflammatory cell infiltration, pulmonary hemorrhage, and vascular congestion. Original magnification: 200×. ASCI, acute spinal cord injury.

**Figure 2 diagnostics-15-02648-f002:**
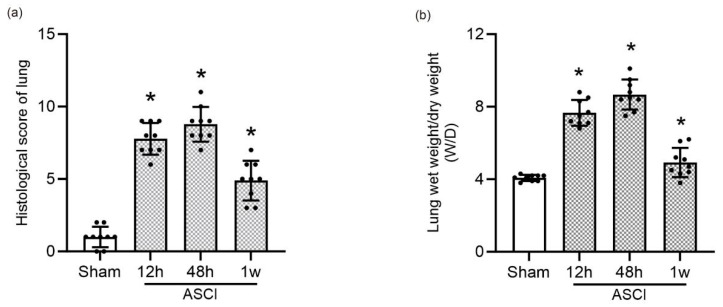
Quantitative Assessment of Lung Injury Following Acute Spinal Cord Injury in Rats. (**a**) Histological injury scores of lung tissue sections at 12 h, 48 h, and 1 week following ASCI or sham surgery (mean ± standard deviation [SD]). (**b**) Pulmonary edema assessment using the wet-to-dry (W/D) lung weight ratio at the same time points (mean ± SD). One-way analysis of variance (ANOVA) followed by the LSD test was used for statistical comparisons. * *p* < 0.05 vs. sham group. ASCI, acute spinal cord injury; W/D, wet-to-dry weight ratio; SD, standard deviation; LSD, least significant difference.

**Figure 3 diagnostics-15-02648-f003:**
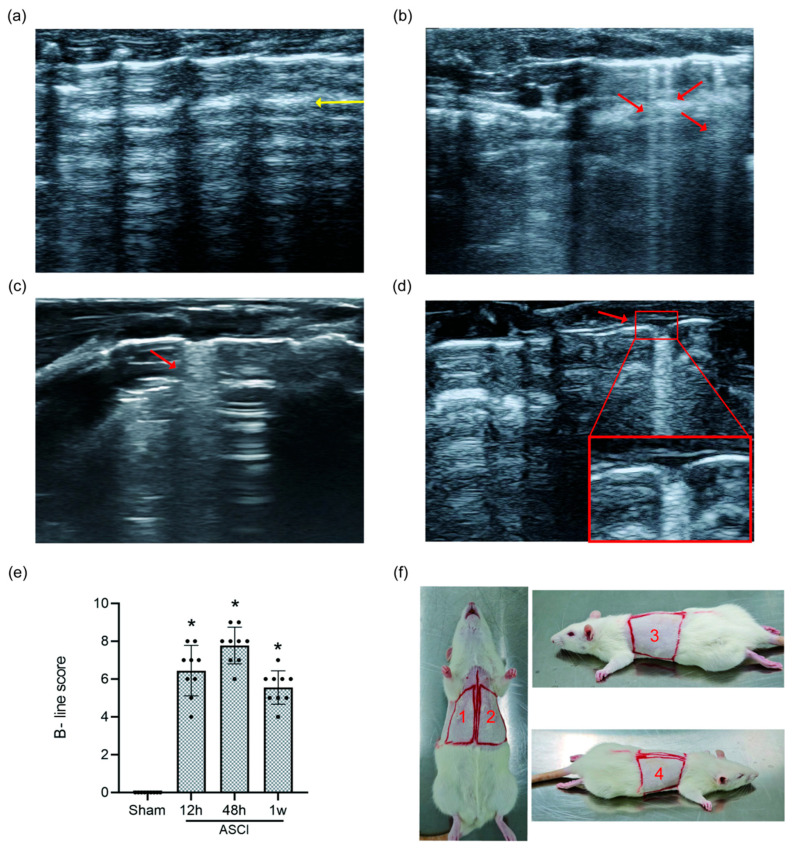
Lung Ultrasound Findings and B-Line Scoring in Rats Following Acute Spinal Cord Injury. Representative LUS images and quantitative scoring at 12 h, 48 h, and 1 week post-injury or sham surgery. (**a**) Sham group showing normal A-lines (yellow arrow). (**b**) At 12 h after ASCI, narrow-based B-lines are observed (red arrow). (**c**) At 48 h, B-lines become confluent with widened bases (red arrow). (**d**) At 1 week, confluent B-lines persist along with moth-eaten pleural changes (red arrow). (**e**) B-line scores (mean ± standard deviation) over time. One-way analysis of variance (ANOVA) followed by the LSD test was used for group comparisons. (**f**) Scanning regions for LUS: the thoracic surface was divided into four regions across the dorsal and ventral chest wall. * *p* < 0.05 vs. sham group. ASCI, acute spinal cord injury; LUS, lung ultrasound; LSD, least significant difference.

**Figure 4 diagnostics-15-02648-f004:**
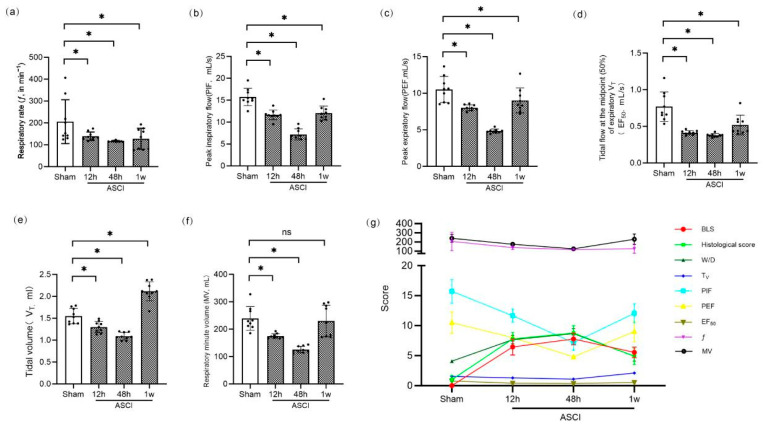
Pulmonary Function and Correlation Analysis in Rats Following Acute Spinal Cord Injury. Pulmonary function parameters and correlation analysis at 12 h, 48 h, and 1 week after ASCI or sham surgery. (**a**) Respiratory frequency (ƒ, breaths/min). (**b**) Peak inspiratory flow (PIF, mL/s). (**c**) Peak expiratory flow (PEF, mL/s). (**d**) Tidal mid-expiratory flow (EF_50_, mL/s). (**e**) Tidal volume (V_T_, mL). (**f**) Minute volume (MV, mL/min). Data are presented as mean ± standard deviation (SD). (**g**) Correlation between BLS and physiological and pathological indices. BLS was negatively correlated with ƒ, V_T_, MV, PIF, PEF, and EF_50_, and positively correlated with histological injury score and W/D lung weight ratio. One-way analysis of variance (ANOVA) followed by the LSD test was used for group comparisons. * *p* < 0.05 vs. sham group; ns, not significant. ASCI, acute spinal cord injury; ƒ, respiratory frequency; V_T_, tidal volume; MV, minute volume; PIF, peak inspiratory flow; PEF, peak expiratory flow; EF_50_, mid-expiratory tidal flow at 50%; BLS, B-line score; W/D, wet-to-dry weight ratio; SD, standard deviation; LSD, least significant difference.

## Data Availability

The datasets generated and/or analyzed during the current study are available from the corresponding author upon reasonable request.

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
