# Peer review of "Lung Ultrasound Assessment of Lung Injury Following Acute Spinal Cord Injury in Rats"

_diagnostics, 2025, doi:10.3390/diagnostics15202648_

Round 1
Reviewer 1 Report (New Reviewer)
Comments and Suggestions for Authors
Dear Authors,
It is my pleasure to review your study.
The article titled "Lung Ultrasound in Lung Injury Following Acute Spinal Cord Injury in Rat" raises an interesting topic, however there are a few minor issues that require correction.
1.In the introduction, the dose of ionizing radiation could be mentioned.
2.The date of approval by the animal bioethics committee should be added.
3.Was the same ultrasound imaging regimen used for all rats?
4.In section "2.4. Histopathological examination" please add a reference to the technique described.
5.A reference should be added in section "2.5. Lung function tests".
6.The image of the lungs in Figure 1 could be of better quality.
7.I don't fully understand the statement "Informed Consent Statement: Informed consent was obtained from all subjects involved in the study." since the study was conducted on rats.
In my opinion, the article is well-prepared, in accordance with the journal's guidelines. The methodology is clear and well-described. The study's conclusions are clear. However, there are minor comments that require correction.
Author Response
Comments 1:In the introduction, the dose of ionizing radiation could be mentioned.
Response 1:We have added relevant research reports on the effects of ionizing radiation in the Introduction section (highlighted in red).
Comments 2:The date of approval by the animal bioethics committee should be added.
Response 2:The date of approval by the Animal Bioethics Committee is included in our protocol number (protocol number: CHCMU-IACUC20240131008, approval date: January 31, 2024).
Comments 3:Was the same ultrasound imaging regimen used for all rats?
Response 3:All rats were treated using the same ultrasound protocol as described in the Materials and Methods section.
Comments 4:In section "2.4. Histopathological examination" please add a reference to the technique described.
Response 4:We have revised and added references (highlighted in red).
Comments 5:.A reference should be added in section "2.5. Lung function tests".
Response 5:We have revised and added references (highlighted in red).
Comments 6:The image of the lungs in Figure 1 could be of better quality.
Response 6:We have revised and provided higher-resolution images.
Comments 7:I don't fully understand the statement "Informed Consent Statement: Informed consent was obtained from all subjects involved in the study." since the study was conducted on rats.
Response 7:This study was conducted using experimental animals and did not involve clinical patients. We have deleted the corresponding content.
Reviewer 2 Report (New Reviewer)
Comments and Suggestions for Authors
Lung complications are one of the emergent conditions in patients with acute spinal cord injury. The trial of ultrasound to evaluate pulmonary function was valuable and interesting. I have some questions.
First, what is the reason for the reduction in pulmonary function after a T10 level injury? Is it caused by progressive weakening of the intercostal and abdominal muscles?
Second, the authors used two ultrasound systems to evaluate lung motion in animal models. It would be helpful to present the image of lung evaluation in a rat in the article. Are there any imaging quality differences?
Author Response
Comments 1:what is the reason for the reduction in pulmonary function after a T10 level injury? Is it caused by progressive weakening of the intercostal and abdominal muscles?
Response 1:Thank you for raising this insightful question. The T10 injury does not affect the nerves innervating the diaphragm and intercostal muscles, and therefore does not lead to paralysis of these muscles. Consequently, the underlying mechanism of lung injury is likely to be complex. Current evidence suggests that it may be associated with an inflammatory storm and could also be modulated via the spinal-lung axis. This mechanism is indeed intricate. In our previous studies, MD Chu and colleagues have dynamically described the progression of lung injury and its potential underlying mechanisms [1-2]. We sincerely acknowledge that further investigation is needed, and our team is actively exploring this question.
- Chu, R.; Wang, J.; Bi, Y.; Nan, G. The kinetics of autophagy in the lung following acute spinal cord injury in rats. Spine J 2018, 18, 845-856.
- Chu, R.; Wang, N.; Bi, Y.; Nan, G. Rapamycin prevents lung injury related to acute spinal cord injury in rats. Sci. Rep. 2023, 13, 10674.
Comments 2:the authors used two ultrasound systems to evaluate lung motion in animal models. It would be helpful to present the image of lung evaluation in a rat in the article. Are there any imaging quality differences?
Response 2:We sincerely appreciate your valuable comments. Regarding the concern you raised about the use of two different ultrasound systems, we would like to provide clarification. As you correctly noted, subtle differences in image quality may exist between different imaging devices. In our study, the first ultrasound system was actively engaged in clinical diagnostic procedures during a critical phase of the experiment. To ensure continuity of our research, we therefore utilized an alternative clinically available ultrasound system.
We would like to emphasize that both ultrasound systems employed in this study are standard devices routinely used in our hospital’s ultrasound department for clinical diagnosis. Both systems provide clear and stable imaging quality, fully meeting our requirements for accurate identification and evaluation of key pulmonary ultrasound features, including A-lines, B-lines, pleural line, lung sliding, and pulmonary consolidation. Consequently, despite differences in brand and model, the images obtained from both systems are diagnostically reliable and do not compromise the accuracy of our dynamic assessment of lung function in this study.
Furthermore, we greatly appreciate your suggestion to include representative rat lung ultrasound images in the manuscript, which indeed enhances the visual clarity of our findings. We have now incorporated representative images into the revised manuscript.
Reviewer 3 Report (New Reviewer)
Comments and Suggestions for Authors
Congratulations to you on your brilliant study, using a non-invasive, non-radioactive and effective method for evaluating and follow-up of the lung injury condition after acute spinal cord injury in animal experiment. I hope and I think it can be implemented in future clinical practice. The entire manuscript was well written in acceptable language and layout. The article can be considered for publication if you address my following comment.
I suggest you should make a table in which the ultrasound findings correspond to the severity of lung injuries.
Author Response
Comments 1:suggest make a table in which the ultrasound findings correspond to the severity of lung injuries.
Response 1:We greatly appreciate this valuable suggestion, which we also consider important as it significantly enhances the readability of our manuscript. To comprehensively reflect the status of lung injury, we have provided multiple types of data from both pathological and physiological perspectives. To better illustrate the correlation of these data with lung ultrasound findings, we employed the visualization approach shown in Fig. 4g. This allows readers to intuitively observe both positive and negative correlations, while also providing access to the specific numerical values.
This manuscript is a resubmission of an earlier submission. The following is a list of the peer review reports and author responses from that submission.
Round 1
Reviewer 1 Report
Comments and Suggestions for Authors
- Abstract: a) Methods: What is the level of the spinal cord injury? It affects the severity of pulmonary injury.
- Introduction: The patterns of pulmonary injury following spinal cord injury should be included in a brief paragraph.
- Materials and Methods: a) Animals: The level and mechanism of spinal injury, along with the age of the rats, should be included.
- Results: The primary issue of the study is that neither the level nor the mechanism of cord injury was evaluated. Both parameters should be analyzed alongside LUS findings.
- Discussion: A paragraph discussing the usefulness of Lung Ultrasound (LUS) in acute pulmonary issues should be included.
Reviewer 2 Report
Comments and Suggestions for Authors
1-Lung ultrasound (LUS) is known to visualize peripheral lung tissues effectively. However, was there any attempt to detect deep parenchymal injury? If not, this could be considered a limitation of the study.
2-Among the regions scanned during LUS (anterior, posterior, inferior, etc.), was there a specific zone where lung injury was more pronounced? If so, was this confirmed histopathologically? Discussing such correlations would enhance the causal interpretation of the results and provide clinical relevance.
3-In human clinical practice, bedside lung ultrasound may not be as easy to perform as in rats, particularly considering the risk of spinal trauma. This could limit the feasibility of this approach in patients with acute spinal cord injury.
4-The study assumes that the lung injury is solely secondary to spinal trauma. However, what is the proposed mechanism of this injury? Further explanation is needed. Additionally, how can the researchers be certain that the lung injury developed as a result of spinal trauma and not from other causes during the experimental procedure?
5-Can it be confidently stated that the same degree of spinal cord injury was created in all rats? If varying severities of spinal trauma result in different levels of lung injury, how was standardization achieved across the experimental groups?
6-Is there any operator dependency associated with the LUS procedure in this model? If so, how was inter-operator variability minimized or controlled?
7-Is the B-line scoring method a standard approach in lung ultrasound? Are there alternative scoring systems? It would be useful to discuss the advantages and limitations of the method used in this study.
8- Although histopathology was used as the reference standard in this study, the inclusion of an additional imaging modality such as CT might have facilitated a more robust comparison and validation of LUS findings.
Reviewer 3 Report
Comments and Suggestions for Authors
General comments to the authors – The current study investigates whether B-mode ultrasound scans of the lung can be used to accurately detect pulmonary injury following acute spinal cord injury (ASCI). The ability of ultrasound to assess changes resulting from extravascular lung water is well-established and has been previously used to predict lung injury following blunt force trauma to the thorax. This is the first study to apply this method to ASCI-related lung injury, identifying correlations between a modified B-line scoring (BLS) system and gross, pathologic, and functional changes after injury in a validated rat model of ASCI. The clinical utility of these findings is unclear, and the manuscript would benefit from revisions to the methodology and results text to ensure replicability and strengthen conclusions. Some specific suggestions are given below.
Specific comments –
Page 3, lines 93 – 94: How were the two ultrasound systems distributed across rats and sonographers? Were they randomly assigned or did one sonographer exclusively use one system? Please clarify.
Page 3, lines 100 – 104: Additional details are needed to describe the B-line scoring methodology. Was each scanning region scored and summed? What is the maximum score? How were discrepancies in scores between sonographers handled? These should be addressed in the manuscript text.
Page 3, lines 116 – 117: What was the histological scoring system employed in this study? Were histological scores averaged between the two pathologists? Please add these details to the text.
Page 3, lines 130 – 133: Comparisons for statistical significance were made between sham and ASCI groups at each time point, but figures throughout the manuscript show only one time point for sham rats. All figures and figure captions should be updated to make clear what is being plotted or to display all sham time points.
Page 5, lines 164 – 168: Quantitative data of histological score and wet-to-dry lung weight ratios are missing from the manuscript. The mean ± standard deviation for each group, as well as their respective p-values, should be added, possibly in the form of a table.
Page 6, lines 185 – 188: Quantitative data of the B-line scores are missing and should be added. It would also be beneficial to compare B-line scores between sonographers and across scanning regions.
Page 7, lines 204 – 209: Same comment as above. Quantitative data of the pulmonary function parameters are missing and should be added for all groups.
Page 7, Figure 4: Due to the large differences in data values and the low resolution of the figure, it is difficult to interpret the trends over time for some of the parameters (e.g., tidal volume and wet-to-dry ratio). Consider revising.
Discussion: No insight as to how the findings from this study might be used to inform clinical treatment is provided. Do the authors envision using lung ultrasound as a one-time confirmation of pulmonary injury or to guide treatment over time? Would the B-line score be used as a sole indicator of injury or combined with measures of pulmonary function? Please describe.
Reviewer 4 Report
Comments and Suggestions for Authors
This study evaluates lung ultrasound (LUS) for pulmonary injury following acute spinal cord injury (ASCI) in rats. There are multiple issues that need to be addressed:
- One major issue of his study and its design is animal model generalizaility. While well-conducted in rats, extrapolation to human ASCI patients is highly speculative. This limitation is perhaps not emphasized enough in the conclusions or abstract. In addition, The model uses only female rats, which may introduce sex-specific bias—unacknowledged in discussion.
- Inter-observer reliability is not reported for BLS scoring despite two independent observers performing LUS. This is a significant omission in a semi-quantitative study.
- The modified BLS system is used without any prior validation, raising concerns over scoring reproducibility and its interpretive value.
- Although the authors refer to hyperimpact severity, the exact criteria for injury severity, variability, and consistency in injury delivery are not well-explained.
- There is no mention of a sample size/power calculation, even though the groups are relatively small (n=9 per time point). This raises the concern of underpowered results.
- Multiple correlations were performed, but no correction for multiple comparisons (e.g., Bonferroni or FDR) was applied.
- The data variability (SD) in some plots (e.g., lung function) seems quite high, but no justification or analysis of inter-subject variation is provided.
- Graphs often lack individual data points (e.g., scatter or violin plots), which would have been more informative given the small n.
- Although the study provides proof-of-concept, no LUS imaging criteria for fibrosis are objectively defined. Moth-eaten appearance is not quantified in angle, echogenicity, or thickness as cited in literature (e.g., Zhou et al.).
- Functional parameters start to diverge from BLS by 1 week, suggesting the score may not capture recovery, and this concern is not thoroughly analyzed.
- No attempt to relate ultrasound findings to clinical management decisions (e.g., threshold for ventilation) is made.
- No ultrasound cine loops or supplementary dynamic clips are provided—essential in LUS studies for peer review and interpretation.
- Figure 3e shows significant differences in BLS, but subjective scoring systems should be accompanied by inter-rater agreement statistics (e.g., ICC or Cohen’s kappa).
- Images showing pleural moth-eaten changes would benefit from zoomed-in or magnified versions for better clarity and objective appreciation.
Comments on the Quality of English Language
- Several sentences are too long or repetitive. Please revise.